Utilizing spent mushroom substrate for rearing black soldier fly (Hermetia illucens) larvae: enhancing fertilizer efficiency and improving animal feed quality for sustainable agriculture

Kanjanarat Kanokkan 1
Laksanawimol Parichart 2
Lersawhanwaree Jittawadee 2
Khan Sarayut 3
Thancharoen Anchana koybio@gmail.com 4 5
1 Faculty of Education, Bansomdejchaopraya Rajabhat University , Bangkok , Thailand
2 Faculty of Science, Chandrakasem Rajabhat University , Bangkok , Thailand
3 Faculty of Science and Technology, Bansomdejchaopraya Rajabhat University , Bangkok , Thailand
4 Department of Entomology, Kasetsart University , Bangkok , Thailand
5 Research and Lifelong Learning Center for Urban and Environmental Entomology, Kasetsart University Institute for Advanced Studies, Kasetsart University , Bangkok , Thailand
Lazzari Claudio
Electronic publication date: 2025 Jun 19
Publication date: 2025
Volume: 13
Electronic Location ID: e19590
Received 2025 Mar 31; Accepted 2025 May 20
Copyright: ©2025 Kanjanarat et al.
Copyright year: 2025
Copyright holder: Kanjanarat et al.
License: This is an open access article distributed under the terms of the Creative Commons Attribution License, which permits unrestricted use, distribution, reproduction and adaptation in any medium and for any purpose provided that it is properly attributed. For attribution, the original author(s), title, publication source (PeerJ) and either DOI or URL of the article must be cited.
License URL: https://creativecommons.org/licenses/by/4.0/

Keywords: Frass, BSFL decomposition, Fermentation, Agricultural product, Protein, Growth performance, Nutrient contents, Survival, Supplementation

Funding: Thailand Science Research and Innovation (TSRI) FF 2567-11: 4695724 Kasetsart University Research and Development Institute (KURDI) FF(KU) 51.68 This study was financed by the Thailand Science Research and Innovation (TSRI) (Grant No. FF 2567-11: 4695724) and the Kasetsart University Research and Development Institute (KURDI) (Grant No. FF(KU) 51.68). The funders had no role in study design, data collection and analysis, decision to publish, or preparation of the manuscript.

==============================
Spent mushroom substrate (SMS), a byproduct of mushroom cultivation, has high potential as a substrate for rearing animals, including black soldier fly larvae (BSFL). However, due to its low nutrient content, mixing it with other organic waste is necessary to enhance its efficiency and effectiveness. We evaluated three types of SMS—Agrocybe cylindracea, Lentinus polychrous, and Pleurotus pulmonarius—supplemented with mixed vegetables at three different levels (0% (VEG or control), 15%, and 30% (w/w)) and subjected to two pre-treatment processes (non-fermented and fermented). The SMS from three different mushroom types did not differ much in their influences upon BSFL growth performance at a 15% (w/w) SMS supplement, and were similar to VEG. Increasing the SMS level to 30% resulted in smaller larval, prepupal, and pupal weights. Self-fermentation of SMS for two months neither significantly altered its nutrient composition nor affected larval growth. However, it resulted in a better nutrient quality of the BSFL biomass and its frass, which was associated with the nutrient composition in SMS. Accordingly, BSFL decomposition significantly enhanced the efficiency of low-nutrient substrates, resulting in a 6- to 10-fold increased protein content in the larval biomass and higher total nitrogen, phosphorous, and potassium (NPK) levels in their frass compared to raw SMS. Nevertheless, further optimizing the substrate formulation would likely enhance the benefits from available waste materials and so support sustainable agriculture.

Introduction

Spent mushroom substrate (SMS), the residual substrate from mushroom cultivation after their harvesting, is a type of agricultural waste that still contains a high level of organic and mineral matter from the unutilized substrate (Kwak, Jung & Kim, 2008). The mushroom substrates used for the growth of different mushroom types vary in composition but are mostly composed of lignocellulose-rich agricultural wastes, such as paddy straw, sawdust, and palm kernel cake (Mohd Hanafi et al., 2018). Therefore, the chemical composition of SMS, specifically sawdust-based SMS for Pleurotus eryngii, exhibits high amounts of neutral and acid detergent fibers, cellulose, hemicellulose, and lignin (Kwak, Jung & Kim, 2008). In addition, SMS may contain unused nutrients, fungal residue, and lignocellulosic enzymes, including phenolic compounds resulting from lignin degradation (Phan & Sabaratnam, 2012).

Currently, global mushroom consumption has reached 34 million tons per year and is continuing to increase (Grimm, Kuenz & Rahmann, 2021). Approximately five kg of SMS waste is produced for every one kg of mushrooms (Finney et al., 2009). Consequently, a substantial amount of SMS is discarded from mushroom farms, particularly those in large-scale production industries, which can cause environmental problems if it is managed by burning or disposal (Mohd Hanafi et al., 2018). To address this issue, the utilization of SMS waste has been widely studied to minimize its impact. For instance, although SMS has a low nutrient composition, which limits its use as an animal feedstock, it can be modified and supplemented in feed to help reduce the cost (Mohd Hanafi et al., 2018). Moreover, it can be used as a biofertilizer and soil amendment (Leong et al., 2022; Mohd Hanafi et al., 2018), as fuel pellets (Alves et al., 2021), and even recycled for mushroom cultivation (Cunha Zied et al., 2020; Rinker, 2017). Additionally, it has been used as a substrate for insect cultivation, such as by mixing with rice bran for mealworms (Li et al., 2020) or with food waste or chicken feed for black soldier fly larvae (BSFL) (Nayak & Klüber, 2025; Nayak, Rühl & Klüber, 2024). However, further research is still needed to explore cost-effective applications of SMS with high benefits.

BSF, Hermetia illucens (L.) (Diptera: Stratiomyidae), is a saprophagous species capable of breaking down various types of organic materials and is, therefore, widely recognized in organic waste management (Gold et al., 2018; Liu, Wang & Yao, 2019). During digestion, BSFL produce antimicrobial peptides, and their gut-associated bacteria suppress pathogenic bacteria (Elhag et al., 2022). Due to their ability to suppress pathogens, BSFL have attracted interest in transforming unhygienic or chemically contaminated waste into higher-grade products (Tepper et al., 2024). Through waste conversion, BSFL produce a high-nutrient larval biomass, which has been used as an alternative protein source for animal feed, including for poultry, pigs, and fish (Abd El-Hack et al., 2020; Nairuti et al., 2021; Veldkamp & Vernooij, 2021). The byproduct of the BSFL decomposition process is frass, a nutrient-rich biofertilizer containing approximately 37% carbon (C), 3% nitrogen (N), 1–5% phosphorus (P), and 0.5–4.1% potassium (K) by wet weight (w/w) (Lopes, Yong & Lalander, 2022). Recently, commercial BSF farms have emerged worldwide for waste management, aligning with the circular economy principles of the United Nations’ Sustainable Development Goals and the commercialization of BSF products.

Although BSFL have the ability to digest a wide variety of wastes, including fruits, vegetables, animal proteins, manure, and high-fiber materials, their growth is limited on substrates with a high levels of moisture (Frooninckx et al., 2024; Laksanawimol, Anukun & Thancharoen, 2024), salinity (Cho et al., 2020), and fat (Klammsteiner et al., 2021), or containing certain chemical compounds (Siddiqui et al., 2023). The use of SMS or sawdust as a supplement in BSFL substrates has been employed to regulate the moisture levels and create more optimal conditions for BSFL growth (Laksanawimol, Anukun & Thancharoen, 2024; Li et al., 2021; Nayak & Klüber, 2025; Nayak, Rühl & Klüber, 2024). According to Nayak, Rühl & Klüber (2024) and Li et al. (2020), 20%–30% (w/w) SMS supplementation provided the highest benefits for BSFL growth performance and protein content in their biomass, while also reducing the feed costs, particularly in the insect industry. However, a better understanding of different mushroom types, pretreatment methods, and the nutrients in the frass is still needed to develop an optimal protocol for SMS usage.

Generally, SMS waste is left for a period of time before utilization, during which its qualities may change; however, information on these changes is lacking. This study aimed to evaluate the potential of SMS supplementation in substrates to enhance the BSFL yield and nutrient composition in both larval biomass and frass. Three different SMS types and two pretreatment steps—self-fermented (fermented) and non-fermented—were examined for their effects on BSFL growth performance. The results can serve as a guideline for optimizing SMS substrate preparation to improve BSFL products. Significantly, they support the circular economy of the mushroom industry in SMS management by converting SMS waste into valuable products.

Materials & Methods

Preparation of BSF eggs and larvae for experiment

The BSF colony has been maintained and bred at Kasetsart University (KU) BSF Farm, Department of Entomology, KU, Bangkok, Thailand, for four years, as previously reported Laksanawimol, Anukun & Thancharoen (2024). In brief, the adult flies were housed in a mesh chamber (4 m (L) × 3 m (W) × 3 m (H)). They laid eggs in the gaps of a wooden sheet placed over a three-day fermented vegetable tray, which served as the oviposition attractant. One day prior to the beginning of the experiment, the eggs were carefully removed to ensure that only eggs of the same age were used for the experiment the following day. Each batch of 0.5 g of eggs was placed in a small plastic cup and then transferred to a hatching container containing a baby food mixture, which was composed of 1,000 g of minced mixed vegetables and 500 g of rice bran, to produce five-day-old larvae (referred to as 5-DOL). Three hatching trays were prepared to allow for additional selection of larvae of the same size. The research was approved for animal care and use for scientific research by Kasetsart University Institutional Animal Care and Use Committee (ACKU68-AGR-005).

Experimental design

The mushroom media typically used for cultivation of various edible mushrooms vary in composition, harvest cycles, and duration. However, the main composition typically includes 80–90% (w/w) rubber sawdust and 3–4% (w/w) rice bran. Three types of SMS (see Table S1 for composition, harvest cycles, and utilized duration)—derived from cultivation of Agrocybe cylindracea (AC), Lentinus polychrous (LP), and Pleurotus pulmonarius (PP)—were selected for evaluation as substrates for BSFL rearing by mixing with mixed vegetables at three different quantities (0% (VEG or control), 15%, and 30% (w/w)) and two pre-treatment processes: non-fermented (NF) and fermented (F). The sawdust-based SMS formulations are quite similar, except for the addition of tapioca starch in PP and cornmeal in AC as additional carbohydrate sources. The proximate analysis of all tested substrates is shown in Table 1. Each substrate test was performed with three replicates, with an additional replicate designated for substrate moisture measurement every 3 d (Ponpe Moisture Analyzer, PONPE 400MB-1). The mixed vegetables, comprised of cabbage, romaine lettuce, tomato, and lettuce sourced from the Royal Project Foundation, were cleaned and minced before use.

Table 1 Proximate analysis (as % dry matter) of all tested substrates with varying SMS, quantities, and fermentation process.

Substrates	Moisture (%)	Crude protein (%)	Crude fat (%)	Crude fiber (%)	Ash (%)	Total carbohydrate (%)	Soluble carbohydrate (%)	
VEG	95.65	19.02	3.12	20.53	10.22	67.65	47.14	
15% LP-NF	85.73	5.07	1.26	33.65	21.72	71.96	38.31	
15% LP-F	85.91	4.55	1.74	24.57	32.44	61.27	36.70	
30% LP-NF	73.13	3.54	1.70	31.43	25.13	69.64	38.21	
30% LP-F	82.33	4.33	1.19	20.20	36.73	57.76	37.56	
15% PP-NF	90.46	6.69	2.31	36.43	13.43	77.57	41.14	
15% PP-F	90.22	7.78	1.24	31.99	31.99	70.74	38.79	
30% PP-NF	87.41	3.71	1.90	38.39	15.34	79.16	40.77	
30% PP-F	87.73	6.35	1.46	33.12	20.44	71.76	38.64	
15% AC-NF	90.01	5.65	2.46	29.03	13.11	78.79	54.26	
15% AC-F	88.07	5.65	3.32	30.31	13.98	77.05	46.75	
30% AC-NF	85.36	3.80	1.27	31.27	14.06	80.88	49.61	
30% AC-F	81.92	3.29	3.09	30.57	15.65	77.98	47.41	
Notes.

AC Agrocybe cylindracea

F fermented

LP Lentinus polychrous

NF non-fermented

PP Pleurotus pulmonariu

VEG control

Total carbohydrates (%) = 100 - (moisture + crude protein + crude fat + ash).

Soluble carbohydrates (%) = 100 - (moisture + crude protein + crude fat + ash + crude fiber).

Each SMS from bag cultivation was first prepared as small pieces (particle size < three mm) through a mincing process, and then either immediately used in its fresh state (NF; 70–90% moisture) or self-fermented (F) through aerobic decomposition for two months in containers (140 cm (L) × 40 cm (W) × 30 cm (H)) placed in a well-ventilated, shaded area. The fermented substrate was turned over daily to enhance aeration. The self-fermentation was designed to correspond with mushroom farming activities, where growers clear the spent mushroom substrate and let it sit for a while for further utilization.

The experiment was divided into two trials to evaluate different parameters.

Trial 1 was a small-scale study to collect data on BSFL growth, BSFL yield, and the frass nutritional analysis. While Trial 1 was suitable for comprehensive data collection, it did not produce a sufficient quantity of BSFL for proximate analysis. The diet trials were conducted in 2-liter plastic containers covered with a fine cloth to exclude other insects while providing aeration. The tested substrates were prepared for a duration of 14 days at a feed rate of 200 mg per larva per day (Laksanawimol, Anukun & Thancharoen, 2024; Nyakeri et al., 2019), with a total diet of 840 g in each replicate. Next, 300 5-DOL (0.0504 ± 0.0101 g each) were gently handled using soft forceps and placed into each replicate of the respective test substrate.

Trial 2 was a medium-scale study using 10 kg of substrate. The top three tested substrates (15% (w/w) SMS, both F and NF) were selected and used in larger containers to produce BSFL biomass. The experiment was conducted in a plastic tray with dimensions of 60 cm (L) × 40 cm (W) × 15 cm (H), and covered with fine cloth. A total of 10 kg of substrate and 0.1 g of BSF eggs (at approximately the same feed rate as Trial 1) were placed in the tray simultaneously.

Growth performance, survival rate, and larval duration of BSFL

During the testing, BSFL developed from larvae to prepupae and subsequently to pupae. The development of BSFL was tracked by weighing each of 60 randomly selected larvae, prepupae, and pupae from each treatment across all replicates (20 individuals per replicate) using an analytical balance (OHAUS Pioneer PA214). The larval stage was sampled as creamy-white larvae when they had reached 40% prepupal appearance, while prepupae were collected as larvae with a black cuticle when 80% of them had reached the prepupal stage (Laksanawimol, Anukun & Thancharoen, 2024). All weighed larval samples were returned to their original experimental containers to continue feeding. The daily appearance of prepupae were counted, weighed, and transferred to pupation containers. The summary of prepupal weight in each replicate was used as the BSFL yield. Pupal weight was recorded once they developed into pupae.

The mortality rate (MR) was assessed by comparing the number of prepupae that successfully developed in each replicate to the initial number of 5-DOL larvae (N = 300). The MR was calculated using Eq. (1): (1) MR%=number of 5-DOL−total number of prepupae/number of 5-DOL×100.

The duration of larval development was interpreted from the time between egg hatching to prepupae. Thus, the daily number of prepupae was converted to the larval duration.

Biochemical analysis of BSFL and substrates

The proximate analysis of all tested substrates and BSFL were analyzed at the Animal Nutrition Laboratory, Department of Animal Science, KU following AOAC (Horwitz, 2010). The moisture, crude protein, crude lipid, crude fiber, and ash contents were all calculated on a dry mass basis. The % total carbohydrate and % soluble carbohydrate were calculated using Eqs. (2) and (3), respectively: (2) Total carbohydrates=100−moisture+crude protein+crude fat+ash

(3) Soluble carbohydrates=100−moisture+crude protein+crude fat+ash+crude fiber.

Nutrient contents of BSFL frass

After all the BSFL samples had developed into pupae, the remaining frass was analyzed for its nutrition contents, including organic carbon (OC), total nitrogen (N), total phosphate (P2O5), total potassium (K2O), and the C/N ratio. The analysis was carried out at the Department of Soil Science, KU using standard procedures described by the Department of Agriculture. The total N content was measured using the Kjeldahl method (Horwitz, 2000). For P2O5 and K2O analysis, the frass sample was first digested in HNO3-HClO4 solution. The P2O5 concentration in the digest was then measured spectrophotometrically using the molybdovanadate method (Horwitz & Latimer, 2005), while the K2O content was analyzed using atomic absorption spectrometry (Horwitz, 2000). The OC was quantified using the Walkley and Black titration method (Walkley & Black, 1934).

Data analysis

Statistical analysis was conducted on the data obtained for the mean weight of BSF (larvae, prepupae, and pupae), BSFL yield, and larval duration. In order to ascertain whether there were substantial variations among the substrate groups, a one-way analysis of variance (ANOVA) was implemented. Fisher’s least significant difference (LSD) test was implemented for multiple comparisons. All statistical analyses were conducted using SPSS, version 14 (SPSS for Windows, Chicago: SPSS Inc.). Statistical significance was defined as a P-value of less than 0.05.

Results

Larval duration

The BSFL reared on the control VEG substrate had an average development duration of 16.4 ± 2.8 d, which was not significantly different from that of the 30% PP-NF treatment. Surprisingly, the 30% AC-NF treatment had the shortest average development duration (15.9 ± 3.3 days), while the other treatments took longer (F = 37.190, df = 12, P <  0.001) (Fig. 1). The quantity of SMS did not affect the larval duration; larvae reared on fermented 15% (w/w) SMS from AC, LP, and PP took slightly longer to develop than those on 30% (w/w) SMS from LP. The NF SMS resulted in a shorter larval duration compared to its fermented counterparts, except for the 30% LP-NF treatment.

Figure 1 Larval duration (d) of BSFL reared on various substrates.

The white and dark markers represent the short-range and long length group of larval duration, respectively. Different quantities of SMS are indicated by white and gray shaded areas. Data are shown as the mean ± 1 standard deviation (SD), derived from three replicates. Different letters indicate significant differences among substrate treatments (P < 0.05).

The BSFL reared on SMS-supplemented substrates developed into prepupae a few days earlier than those in the VEG treatment. Asynchronous development to the prepupal stage was clearly observed in SMS-supplemented substrates, resulting in a prolonged duration of prepupal occurrence compared to the VEG treatment (Fig. 2). Since the experiment was performed for only 28 d, some BSFL had not fully developed into the prepupal stage. In most 30% (w/w) SMS-supplemented substrates, 12–35% of BSFL remained undeveloped.

Figure 2 Stacked area chart showing the appearance of prepupae reared on various substrates.

Data are shown as the mean derived from three replicates. Different letters indicate significant differences among substrate treatments (P < 0.05).

Larval survival

The overall survival rate of BSFL across all treatments was very high (99.3 ± 2.9%). All substrates containing SMS exhibited a 100% survival rate except for a replicate of the 30% LP-NF (96.7%); whereas the VEG substrate showed a lower survival rate of 91.4 ± 7.4% (F = 3.713, df = 12, P = 0.002).

Growth performance

Substrates supplemented with higher quantities of SMS (w/w) resulted in lower weights at all three developmental stages (larva, prepupa, and pupa) of BSF (Figs. 3A–3C) (F = 25.706, df = 12, P < 0.001 in larvae; F = 68.894, df = 12, P < 0.001 in prepupae; and F = 43.368, df = 12, P <  0.001 in pupae) compared to VEG. All 30% (w/w) SMS substrates led to lower BSFL weights, particularly the AC substrates. Interestingly, in the larval stage, the weights of BSFL in the 15% PP-NF and 15% AC-F treatments were higher and not significantly different from the control (VEG) group (0.1679 ± 0.0299 g and 0.1643 ± 0.0364 g, respectively). However, none of the SMS-supplemented substrates resulted in larval weights comparable to the VEG control in the prepupal and pupal stages.

Figure 3 Mean (±1 SD) weight (g) of BSF (A) larvae, (B) prepupae, and (C) pupae when reared on various substrates.

The dark and white markers represent the short-range and long length group of larval duration, respectively. Different quantities of SMS are indicated by white and gray shaded areas. Data are shown as the mean ± 1 SD, derived from three replicates. Different letters indicate significant differences among substrate treatments (P < 0.05).

The BSFL yield, summarized as the average wet weight of the prepupal stage for each replicate, displayed the same trend as the individual prepupal weight (Fig. 4). The VEG treatment resulted in the highest BSF yield compare to the 15% and 30% SMS-supplemented substrates. However, most 15% and some 30% SMS-supplemented substrates did not differ significantly from the VEG group (F = 2.225, df = 12, P = 0.043). The BSFL cultured on 30% AC substrates showed the lowest yields, regardless of fermentation.

Figure 4 Mean (±1 SD) BSFL yield (g) reared on various substrates.

The white and dark markers represent the high-range and low-range group of larval duration, respectively. Different quantities of SMS are indicated by white and gray shaded areas. Different letters indicate significant differences among substrate treatments (P < 0.05).

Nutrient contents of BSFL products

The BSFL converted low-nutritional substrates into high-nutritional biomass. For instance, the protein content of the VEG substrate was nearly doubled from 19% (Table 1) to 43% (Table 2). Similarly, BSFL reared on SMS-supplemented substrates exhibited a 6- to 10-fold increase in protein content, reaching 43.70 ± 1.63%. Moreover, the fat content (% fat) was increased up to 20-fold compared to that in the substrate, as observed in the 15% PP-F treatment.

Table 2 Proximate analysis (as % dry matter) of BSFL yields at the prepupal stage reared on various substrates.

Substrate	Moisture (%)	Crude protein (%)	Crude fat (%)	Crude fiber (%)	Ash (%)	Total carbohydrate (%)	Soluble carbohydrate (%)	
VEG	76.98	42.91	28.67	10.99	12.33	16.09	5.10	
15% LP-NF	75.69	42.82	22.78	9.79	17.93	16.47	6.68	
15% LP-F	72.77	44.91	21.22	9.07	20.01	13.86	4.79	
15% PP-NF	75.42	45.60	21.68	9.19	16.11	16.61	7.42	
15% PP-F	74.18	44.61	24.67	9.37	15.68	15.04	5.67	
15% AC-NF	74.71	41.24	23.44	9.29	19.37	15.95	6.66	
15% AC-F	74.56	43.00	25.11	10.33	16.86	15.03	4.70	
Average	74.90	43.58	23.94	9.72	16.90	15.58	5.86	
SD	1.32	1.51	2.53	0.71	2.58	0.98	1.07	
Notes.

AC Agrocybe cylindracea

F fermented

LP Lentinus polychrous

NF non-fermented

PP Pleurotus pulmonariu

VEG control

Total carbohydrates (%) = 100 - (moisture + crude protein + crude fat + ash).

Soluble carbohydrates (%) = 100 - (moisture + crude protein + crude fat + ash + crude fiber).

Comparison of the BSFL products among the tested treatments revealed variations in the nutritional content. When cultured on 15% (w/w) SMS substrates, the BSFL generally exhibited a higher protein content than the control VEG ones, which had the highest protein and fat percentages. The protein content reached up to 45.60% in 15% PP-NF, compared to 42.91% in VEG, while the fat content was noticeably lower.

Nutrient contents of BSFL frass

Prior to digestion by BSFL, the nutrient content in the tested substrates was low (below the standard for organic fertilizers), except for VEG, which had a high total N content (2.32%) (Table 3). Most 15% (w/w) SMS-supplemented substrates exhibited slightly higher percentages of total N, total P2O5, and total K2O after the self-fermentation process, except for AC. These nutrient levels increased further after BSFL digestion, surpassing the organic fertilizer standard, particularly in VEG. All tested substrates were predominantly converted into high-K biofertilizer. A higher N content was found in the SMS from PP and AC, while that from AC was highest in total P2O5. The fermentation process resulted in varying nutrient content outcomes: SMS from LP and AC showed lower total N and K2O contents but maintained the same total P2O5 levels, whereas that from PP exhibited an increase in all nutrient contents.

Table 3 Nutrient contents of various substrates and frasses after BSFL digestion.

Treatment	Substrate	Frass	
	%OC	%Total N	%Total P2O5	%Total K2O	C/N ratio	%OC	%Total N	%Total P2O5	%Total K2O	C/N ratio	
VEG	47.90	2.32	0.82	4.51	20.70	31.00	2.50	3.93	18.20	12.40	
15% LP-NF	37.00	0.54	0.80	1.31	68.50	33.70	1.15	1.65	4.23	29.20	
15% LP-F	31.50	0.82	1.21	2.12	38.40	29.80	1.05	1.63	3.74	28.30	
15% PP-NF	47.30	0.81	0.74	1.28	58.40	40.00	1.28	1.47	4.86	31.30	
15% PP-F	44.00	1.13	1.12	1.60	38.90	37.70	1.37	1.62	6.02	27.60	
15% AC-NF	51.30	0.75	1.74	0.99	68.50	40.00	1.28	2.15	4.29	31.30	
15% AC-F	47.00	0.74	2.00	0.90	63.50	38.80	1.21	2.26	4.10	32.00	
Notes.

AC Agrocybe cylindracea

F fermented

LP Lentinus polychrous

NF non-fermented

OC organic carbon

PP Pleurotus pulmonariu

VEG control

Discussion

BSFL were able to complete their development successfully on SMS supplemented with mixed vegetables, although high levels of SMS supplementation negatively affected larval growth, resulting in smaller larvae. However, the resulting BSFL biomass and frass showed improved quality.

Several studies have investigated the nutrient content remaining in SMS, which revealed they vary widely. For instance, the crude protein content was found to range from 9.5–37.6%, crude fat from 0.7–8.4%, and crude fiber from 1.87–39.55%, which likely depended on the substrate formulations and mushroom types (Luo & Chen, 2024; Mohd Hanafi et al., 2018). Interestingly, the crude protein content in SMS was shown to increase significantly after use (Li & Wang, 2023; Luo & Chen, 2024), along with the presence of bioactive compounds and cellulolytic enzymes, including the bacteria Enterobacter spp. and Bacillus spp. that provide digestive health benefits to animals (Baptista et al., 2023; Kim et al., 2012). Therefore, supplementing animal feed with SMS has been used for several decades for ruminants, poultry, and fish (Baptista et al., 2023). Similarly, in insect diets, SMS has been supplemented with other high-nutrient ingredients, such as rice bran, wheat bran, food waste, and chicken feed, to enhance the nutritional composition of the substrates and control its moisture content (Li et al., 2020; Li et al., 2021; Li & Wang, 2023; Nayak, Rühl & Klüber, 2024). However, some nitrogen loss was observed through ammonia (NH3) evaporation during the waste decomposition process (Coudron et al., 2024).

Barragan-Fonseca et al. (2019) found that low protein-to-carbohydrate (P:C) ratio substrates influenced BSFL growth, and that a diet containing 17% protein and 55% carbohydrate supported both optimal growth performance and high larval body protein content. They concluded that high carbohydrate concentrations and low to medium protein levels are required to support BSFL growth. According to Luo & Chen (2024) and Mohd Hanafi et al. (2018), the SMS they tested contained a significantly higher crude protein and crude fat content than the sawdust-based substrates used in this study. This was the case in this study even when mixed with VEG, and resulted in significantly lower larval weights (155 mg per larva at 15% (w/w) SMS and 127 mg per larva at 30% SMS). In contrast, previous studies that evaluated SMS supplemented with food waste reported higher larval weights (242 mg and 255 mg per larva for 20% and 30% (w/w) supplementation, respectively; Li et al., 2021) or chicken feed (229 mg and 178 mg per larva for 20% and 40% (w/w) supplementation, respectively; Nayak, Rühl & Klüber, 2024). Thus, the selection of substrates for supplementing SMS is crucial in improving BSFL production. We recommend that substrates with high nutritional contents are a more optimal choice for mixing with SMS to achieve higher BSFL weights.

Fungal pretreatment is a method for digesting lignocellulosic biomass (Manyi-Loh & Lues, 2023). As such, SMS appears to be a byproduct that has already undergone this process during mushroom cultivation as lignocellulosic enzymes such as laccase, xylanase, lignin peroxidase, cellulase, and hemicellulose, can be found in the SMS (Sabaratnam et al., 2023). Pérez-Chávez, Alberti & Albertó (2022) reported that the enzyme activity in SMS depends on the type of mushroom and how well it sticks to the substrate. These enzymes can enhance polysaccharide digestibility, facilitating the breakdown of wood fibers in the SMS (Mori et al., 2023), which in turn provides a carbohydrate source for BSFL growth. This is consistent with the observed high weight of BSFL reared in 15% (w/w) SMS, similar to VEG, in this study. Thus, SMS might be a dry substrate that helps support a better BSFL growth performance compared to the other dry substrates previously evaluated by Laksanawimol, Anukun & Thancharoen (2024), due to its enzyme activities.

Substrate formulations for different mushroom species vary, resulting in different physical, chemical, and biological properties in SMS (Luo & Chen, 2024). Although these are edible mushrooms, not all of them can be reliably used as feed material for insects. For instance, L. edodes SMS was the only one among five tested substrates on which yellow mealworms could survive (36.7%), while the other mushroom types resulted in very low survival rates or complete mortality (Li et al., 2020). On the other hand, BSFL are more flexible to survive and grow on various kinds of mushroom SMS (Li et al., 2021; Nayak, Rühl & Klüber, 2024). These results are consistent with this study, which found that BSFL can be reared on all three tested SMS substrates with a very high survival rate.

The proportion of supplementation is a significant factor influencing the growth performance and developmental duration of BSFL. Higher proportions of supplemented materials tended to result in a lower larval weight and prolonged development, indicating a suboptimal diet (Khaekratoke, Laksanawimol & Thancharoen, 2022; Laksanawimol, Anukun & Thancharoen, 2024). Li et al. (2021) suggested that 20–30% (w/w) SMS in mixed substrates is the optimal proportion for BSFL culture, a finding supported by Nayak, Rühl & Klüber (2024). In contrast, this study did not show the same relationship; BSFL reared on some 30% (w/w) SMS treatments exhibited a short larval duration similar to the control. Prepupae appeared earlier in SMS-containing substrates in each replicate (Fig. 2), suggesting that SMS-mixed substrates may provide more suitable moisture conditions for pupation compared to VEG (see Table S2 for substrate moisture monitoring data).

Pre-treatment of SMS by self-fermentation for two months resulted in variable outcomes depending on the mushroom type. The protein content increased significantly only in PP substrates but did not positively correlate with BSFL growth performance. The results for SMS-NF and SMS-F are likely to be similar. According to Kwak, Jung & Kim (2008), Pleurotus eryngii sawdust-based SMS stored in deep stacks for 1–3 weeks showed no changes in their chemical composition, including the levels of crude protein, hemicellulose, and cellulose. This finding is consistent with the proximate analysis of the tested substrates in the present study, indicating that self-fermentation of raw SMS does not improve its composition. However, mixing SMS with broiler poultry litter (stored for three weeks) or pre-treating it with bacterial inoculation (Lactobacillus plantarum strains) and molasses under anaerobic conditions has been reported to improve the SMS quality (Kim et al., 2016; Kwak, Jung & Kim, 2008).

Rearing substrates directly influence the nutritional composition of BSFL, with higher-protein substrates producing larvae with greater total protein levels. For example, fish waste was reported to yield larvae with a total protein content of 78.8%, whereas those reared on fruit and vegetable waste substrates contained only 12.9% protein (Hopkins et al., 2021). In this study, BSFL reared on vegetable-based substrates supplemented with SMS had a high protein content, ranging from 41% to 46%. Interestingly, this was higher than that reported for larvae reared on spent barley with brewer’s yeast (32%), fish (Oncorhynchus mykiss) with wheat (42%), or restaurant waste (16%) (Hopkins et al., 2021). Likewise, the PP substrate, which had the highest protein content, also produced the highest protein levels in larval biomass, even in the control group. However, a low crude protein substrate can still yield a high protein content in the larval biomass, which is likely to be due to the impact of bioactive compounds, including lignocellulosic enzymes, present in the substrate (Phan & Sabaratnam, 2012).

Currently, BSFL frass production has increased due to the growing interest in BSF farming and its products. The advantages of using frass as a biofertilizer extend beyond providing micro- and macro-nutrients; it also enhances nutrient absorption by plant tissues, supplies biomolecules and microorganisms that promote plant growth, and contains compounds that improve plant tolerance to abiotic stress and pathogens (Poveda, 2021). As mentioned above, the production of high-quality BSF frass has the potential to reduce the use of chemical fertilizers and pesticides, and so contribute to sustainable agriculture. Although SMS can be used as a fertilizer or planting substrate and can enhance microbial diversity and abundance, a partial replacement (25% (w/w)) with chemical fertilizers is still necessary to optimize soil nutrient levels (Chen et al., 2022) while a 75% (w/w) mixture with soil can promote plant germination and growth (Verma, Didwana & Maurya, 2020). Ideally, frass should be naturally enriched with essential nutrients for plants, eliminating the need for supplementation with other materials. This study showed that BSF frass from all the SMS treatments contained significantly higher total N, P2O5, and K2O levels compared to those before BSFL digestion. Notably, a high total P2O5 was detected across all treatments, particularly in VEG. The potassium content in SMS frass was likely influenced by the mixed vegetable supplementation.

Although the NPK levels in frass exceeded the standard quality requirements for organic fertilizers, the C/N ratio of frass from SMS substrates remained above 20. However, the total NPK content was between 6–9%, surpassing 2%. Each mushroom substrate contributed different NPK compositions, reflecting the nutritional variations in the BSFL substrates, as also previously observed (Hobbie, 2005). The PP substrate, which had the highest total N content, also produced frass with the highest N levels, whereas AC was dominant in P2 O5. In contrast, LP frass exhibited the lowest biofertilizer quality. Thus, the frass quality can be standardized by adding high-protein wastes into the BSFL substrate when supplemented with SMS.

In a recent study, BSFL decomposition took approximately 12 d to reach the prepupal, non-feeding stage, and completed the prepupation period around day 21. This rate is comparable to the decomposition rate in composting a biofertilizer with effective microorganisms (Boraste et al., 2009). In contrast, BSFL exhibited a high ability to suppress most pathogenic bacteria (Awasthi et al., 2020; Elhag et al., 2022; Kinney et al., 2022), whereas such contamination is still found in compost, persisting in the soil environment and potentially contaminating food products (Seneviratne et al., 2011).

Although SMS waste can be utilized in various ways, this study suggests its alternative potential application as a BSF rearing substrate. Its benefits can be enhanced for both biofertilizer production and conversion into animal feed. In conclusion, our study provides a guideline for using sawdust-based SMS in BSF rearing to support sustainable agriculture

1. The nutritional value of SMS is generally low; therefore, mixing it with high-nutrient substrates is necessary to produce high-quality biofertilizer and protein-rich feed.

2. A supplementation level of 15–20% (w/w) SMS is optimal, as higher proportions result in reduced larval biomass and longer harvest durations.

3. Fresh SMS and SMS subjected to self-fermentation for less than two months show no significant differences in their nutrient composition. However, anaerobic fermentation with manure or inoculation with cellulolytic bacteria may improve SMS substrate quality.

Conclusions

Generally, SMS consists of low-nutrient components for supporting BSFL growth performance; however, it has the potential to serve as a supplement for BSFL rearing substrates due to both its unused nutrients and its ability to regulate substrate moisture. The nutritional composition of SMS varies by mushroom type, depending on the substrate formulation. Mushroom substrates enriched with protein sources, such as tapioca starch (PP) or cornmeal (AC), exhibit higher nutrient levels.

The nutritional composition of fresh SMS and self-fermented SMS (up to two months) is similar, providing flexibility for farmers in its use. To achieve high BSFL growth performance, SMS should be limited to less than 30% (w/w) in rearing substrates, with 15% (w/w) supplementation yielding better results than 30% (w/w). Additionally, mixing SMS with other substrates influences the nutritional content of both the larval biomass and frass.

Supplemental Information

Supplemental Information 1 Dataset

Supplemental Information 2 Composition, harvest cycles, and utilized duration of the tested mushroom media

AC, Agrocybe cylindracea; LP, Lentinus polychrous; PP, Pleurotus pulmonarius; NF, non-fermented and F, fermented.

Supplemental Information 3 Substrate moisture (%) of various tested substrates measured every three days until day 24

The green and red colors represent the highest and lowest values, respectively, with the gradient between the two colors displaying the moisture trend. VEG, control; AC, Agrocybe cylindracea; LP, Lentinus polychrous; PP, Pleurotus pulmonarius, NF, non-fermented and F, fermented.

The authors thank the Royal Project Foundation for providing the vegetable waste used in these experiments to raise the BSF stock. We also appreciate the support from three mushroom farms for supplying SMS: Bangkachao Farm, Samut Prakan province, for AC; Phantouch Mushroom Farm, Bangkok, for LP; and U-Thong Banana Garden Community Enterprise, Suphan Buri province, for PP.

Additional Information and Declarations

Competing Interests

Author Contributions

Ethics

Data Availability

The authors declare there are no competing interests.

Kanokkan Kanjanarat conceived and designed the experiments, performed the experiments, analyzed the data, prepared figures and/or tables, authored or reviewed drafts of the article, and approved the final draft.

Parichart Laksanawimol conceived and designed the experiments, performed the experiments, analyzed the data, prepared figures and/or tables, authored or reviewed drafts of the article, and approved the final draft.

Jittawadee Lersawhanwaree performed the experiments, analyzed the data, prepared figures and/or tables, and approved the final draft.

Sarayut Khan analyzed the data, authored or reviewed drafts of the article, and approved the final draft.

Anchana Thancharoen conceived and designed the experiments, performed the experiments, analyzed the data, prepared figures and/or tables, authored or reviewed drafts of the article, and approved the final draft.

The following information was supplied relating to ethical approvals (i.e., approving body and any reference numbers):

This animal use protocol has been submitted and reviewed by the Kasetsart University Institutional Animal Care (ACKU68-AGR-005).

The following information was supplied regarding data availability:

The raw data are available in the Supplementary File.

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
