# Peer review of "Utilizing spent mushroom substrate for rearing black soldier fly (Hermetia illucens) larvae: enhancing fertilizer efficiency and improving animal feed quality for sustainable agriculture"

_PeerJ, doi:10.7717/peerj.19590_

## Round 0.1 · original submission · Minor Revisions

Both reviewers have given positive feedback on your manuscript. However, they suggest some minor changes that would improve the manuscript.

Reviewer 1 ·

Basic reporting

Paper submitted by Kanjanarat est al present original results on the use of mushrooms culture byproducts for the rearing of Black soldier fly larvae. As predicted, this very poor substrate does not lead to performance but nevertheless allows a certain bioconversion.
Ms is well writen and clear
Background appears adequate, it cannot be exhaustive because there is a huge amount of publications on BSF substrates.
Structure of the ms is correct
Hypotheses and results are in accordance. However I think that results could be interpreted in additionnal ways as gas emission and dry matter concentration (cited line 214 without result), allowing novel conclusions.
line 184: change 'evaluated' to 'measured'
results: change P=0.000 to P<0.001
Fig 3b, a piece of table is on the graph
Begin discussion with a summary of results
Lines 327 and 328, larvae cannot have either 80% or 13% protein contents, that corresponds to %dry matter and fresh larvae because usually BSFL contain ca 68% water.
Line 364, they are much more empirical studies that conclude that BSFL do not prevent the development of pathogenic bacteria.

Experimental design

Clear and well done
Usually, 'DOL' is Day of Laying, but it is used here as day of eclosion
A positive control could miss, as chicken feed or Gainesville diet, but references to other substrates are cited in discussion, it compensates the absence of control diet in experiments. Here, vegetables are considered as control.

Validity of the findings

Globally clear.
I suggests adding some considerations about a putative loss of nitrogen that may have evaporate during the larval growth.
Moreover, if relative moisture oif substrate, frass and larvae are available, they could be included in results.
Graphs are only raw data, some calculations could be done as global FCR, Nitrogen conversion...
On graphs, indicate the number of samples.

Additional comments

The paper is correct, but could be improved with some additional considerations on the results

Reviewer 2 ·

Basic reporting

This research is beneficial for exploring the potential of using waste as a diet for BSFL.

Experimental design

methods described with sufficient detail.

Validity of the findings

no comment

Additional comments

1) There are some differences in BSFL development after feeding on each type of SMS (AC, LP, and PP). Could you explain What are the main factors?
2) Lines 284–290: What is the byproduct of the lignocellulosic enzyme? Is it involved in carbohydrate breakdown? Please explain how important is it for BSF?
3) What is the optimal nutrient ratio (protein: carbohydrate: lipid) for BSF rearing? Based on the proximate analysis in Table 1, which substrate appears to be the most suitable for BSF? Does this correspond with your results?
4) What does a longer development duration indicate?
5) The table should be self-explanatory. Please add footnotes to explain the abbreviations: AC, LP, PP, F, and NF. Please check for all table.
6) Figure caption: Did you mean the dark and white circular markers? From the figure, it seems that the dark markers represent longer lengths and the white markers represent shorter lengths, which appears to be the opposite of what the caption states. What does the gray shading represent? Is it a different experiment or simply used to make the figure easier to read?

---

## Round 0.2 · Minor Revisions

Thank you very much for the improved version, which takes into account the reviewers' suggestions. However, Reviewer 1 highlights a minor point that deserves your attention.

Reviewer 1 ·

Basic reporting

Authors corrected the ms according to my questions
Some minor points:
in table 2, proteins, fat... are on dry matter, it has to be indicated in caption
and the sum of (Proteins+fat+fibers+ash+carbohydrate) exceeds 100%, does if means that fibers are calculated as carbohydrates-sugars? If yes, indicate it in caption.
In table 3, what is "OC%"?

Experimental design

ok

Validity of the findings

ok

Additional comments

none

Reviewer 2 ·

Basic reporting

The revised manuscript is clearly written and well-structured. The authors have addressed all previous concerns regarding clarity, figure presentation, and tables were modified. All data and supplemental materials are appropriately provided.

Experimental design

No comment

Validity of the findings

The revisions have improved the clarity and reliability of the findings.

Additional comments

I commend the authors for their careful and comprehensive revisions. The manuscript is now suitable for publication. I have no further comments.

---

## Round 0.3 · accepted · Accept

Thank you for improving the manuscript in line with the suggestions of the reviewers.

Reviewer 1 ·

Basic reporting

Authors corrected the ms as I suggested

Experimental design

ok

Validity of the findings

ok

Additional comments

no